# Epigenetic Studies in the Male APP/BIN1/COPS5 Triple-Transgenic Mouse Model of Alzheimer’s Disease

**DOI:** 10.3390/ijms23052446

**Published:** 2022-02-23

**Authors:** Olaia Martínez-Iglesias, Vinogran Naidoo, Iván Carrera, Ramón Cacabelos

**Affiliations:** EuroEspes Biomedical Research Center, International Center of Neuroscience and Genomic Medicine, 15165 Corunna, Spain; neurociencias@euroespes.com (V.N.); biotecnologiasalud@ebiotec.com (I.C.); rcacabelos@euroespes.com (R.C.)

**Keywords:** Alzheimer’s disease, epigenetics, sirtuins, HDACs, 3xTg-AD

## Abstract

Alzheimer’s Disease (AD) is a major health problem worldwide. The lack of efficacy of existing therapies for AD is because of diagnosis at late stages of the disease, limited knowledge of biomarkers, and molecular mechanisms of AD pathology, as well as conventional drugs that are focused on symptomatic rather than mechanistic features of the disease. The connection between epigenetics and AD, however, may be useful for the development of novel therapeutics or diagnostic biomarkers for AD. The aim of this study was to investigate a pathogenic role for epigenetics and other biomarkers in the male APP/BIN1/COPS5 triple-transgenic (3xTg) mouse model of AD. In the APP/BIN1/COPS5 3xTg-AD mouse hippocampus, sirtuin expression and activity decreased, *HDAC3* expression and activity increased, *PSEN1* mRNA levels were unchanged, *PSEN2* and *APOE* expression was reduced, and levels of the pro-inflammatory marker IL-6 increased; levels of pro-inflammatory COX-2 and TNFα and apoptotic (NOS3) markers increased slightly, but these were non-significant. In fixed mouse-brain slices, immunoreactivity for CD11b and β-amyloid immunostaining increased. APP/BIN1/COPS5 3xTg-AD mice are a suitable model for evaluating epigenetic changes in AD, the discovery of new epigenetic-related biomarkers for AD diagnosis, and new epidrugs for the treatment of this neurodegenerative disease.

## 1. Introduction

Neurodegenerative disorders (NDDs) are major public health problems and are usually linked to ageing. Alzheimer’s disease (AD) is the leading cause of dementia (>50%), affecting 45–50 million individuals worldwide [1]; over 58% of these patients live in low- and middle-income countries [2]. The prevalence of all-cause dementia is estimated to rise from 50 million in 2010 to 113 million by 2050 [3,4]. AD affects an estimated 6.2 million Americans aged 65 years and older. Unless medical breakthroughs are made to prevent, halt, or cure AD, this figure could rise to 13.8 million people by 2060. According to official death records, AD was the sixth-leading cause of death in the United States in 2019, and the fifth-leading cause of death among Americans 65 years and older. Fatalities from stroke, heart disease, and HIV declined between 2000 and 2019; however, recorded deaths from AD increased by more than 145% during this same period [5]. This increased trajectory in the number of AD-related fatalities was likely exacerbated in 2020 by the COVID-19 pandemic [5]. Despite significant effort over the past 50 years, no successful therapeutic outcomes have been achieved, and anti-AD drugs are not cost-effective [6]. New biomarkers that allow the early diagnosis and treatment of AD are, therefore, required.

AD is progressive and causes an irreversible loss of memory and cognitive function [1]. The main pathological hallmarks of AD are neuritic plaques and intracellular neurofibrillary tangles, which are related to the accumulation of the amyloid-beta peptide (Aβ) and hyperphosphorylated microtubule-associated tau protein, respectively [7,8]. AD is a complex disorder in which genomic, epigenomic, cerebrovascular, metabolic, and environmental factors are potentially involved [9,10].

Transgenic mouse models of AD are promising tools for understanding the underlying mechanisms in AD [11]. A novel triple-transgenic (3xTg)-AD mouse model was recently generated in which mice overexpress the Swedish mutation of human amyloid precursor protein (APP), bridging integrator 1 (BIN1), and COP9 constitutive photomorphogenic homolog subunit 5 (COPS5) [12,13]. These APP/BIN1/COPS5 3xTg-AD mice exhibit Aβ and tau pathology [11], a high level of anxiety and fear [11], severe neuropathological degeneration [4], and deficits in synaptic plasticity, object recognition, and learning [14]. The combination of *APP*, *Bin1* and *COPS5* genes plays a crucial role in the onset of neurodegenerative AD hallmarks, particularly in the early development of Aβ plaques in the neocortex and hippocampus [15]. APP/BIN1/COPS5 3xTg-AD mice develop extracellular Aβ deposits prior to those observed in the double-transgenic mouse model of AD (APP/PS1); sparse fibrillar deposition is visible in those regions in the 0–1-month-old triple-transgenic newborn mouse cortex and hippocampus. As early as six months of age, APP/BIN1/COPS5 3xTg-AD mice exhibit severe Aβ deposition, a finding that supports the use of this model for studying mechanisms of neurodegeneration in AD. The behavioral and psychological symptoms of dementia commonly presented by patients have also been observed in several transgenic mouse models of AD-related pathology, including APP/PS1, Tg2576, 3xTg-AD, 5xFAD, and APP23 [15]. In the present study, we therefore used APP/BIN1/COPS5 3xTg-AD mice to investigate epigenetic changes that may be associated with AD pathology found in human patients.

Epigenetics is the study of reversible heritable changes in gene expression that occur without changes to the DNA sequence, linking the genome and the environment [16,17,18]. The accumulation of epigenetic alterations throughout the lifespan may lead to cerebrovascular and neurodegenerative disorders [19,20]. Classic epigenetic mechanisms include DNA methylation, chromatin remodeling/histone modifications, and micro RNA (miRNA) regulation [19,20,21], and affect gene expression patterns [22]. Significant research efforts are being directed towards understanding the potential role of epigenetics in AD pathogenesis. Epigenetic changes, including DNA methylation and histone modifications, are implicated in learning and memory and have recently been recognized as prospective targets for AD therapy [23,24]. DNA methylation and demethylation are altered in AD [25]. In AD, there is a decrease in DNA methylation in hippocampal and cerebral cortical cells [25]. Moreover, there is a global reduction in 5mC levels and an increase in 5hmC expression in the brains of aged 3xTg-AD mice compared to wild-type animals [26]. In fact, there are low levels of 5mC in the hippocampus, cerebral cortex, and cerebellum of patients with AD [27,28,29]. In the brains of monozygotic twins discordant for AD, 5mC levels were reduced in several cell types in the AD twin than in the non-demented twin [30]. Alterations in HDAC expression, however, have been reported in AD [25]. Increased HDAC3 expression is associated with decreased memory in the AD mouse brain, whereas the induced loss of HDAC3 expression in the dorsal hippocampus improves memory [31,32,33]. Tau acetylation promotes pathological tau aggregation [34]; SIRT1 deacetylates tau, but SIRT1 expression is decreased in both mouse and human cortex in AD [34,35]. Since neuroepigenetics may play an important role in neurodegenerative processes, the current study investigated whether the APP/BIN1/COPS5 3xTg-AD mouse model is a good tool for studying epigenetic contributions to AD. Here, we tested whether this model aids the discovery of new epigenetic biomarkers for the early detection of AD, such that novel therapeutic strategies that utilize epidrugs may be used to treat AD. We analyzed SIRT and HDAC3 expression and activity, the expression of several AD-related genes (PSEN1, PSEN2, APOE), pro-inflammatory (COX-2, TNF-α, and IL-6), and apoptotic (NOS3) markers, and immunoreactivity for glial fibrillary acidic protein (GFAP), CD11b and β-amyloid. Our findings reveal that the APP/BIN1/COPS5 3xTg-AD mouse model is a useful tool for studying the role of epigenetics in AD pathogenesis.

## 2. Results

### 2.1. APP/BIN1/COPS5 3xTg-AD Mice Strongly Express AD-Related Pathological Hallmarks

The primary degenerative effect in AD is irreversible neuronal deterioration, caused in part by the activation of inflammatory processes that induce apoptotic pathways and subsequent necrosis, resulting in massive neuronal death. Figure 1A–D’ highlights the neurodegenerative process observed in the APP/BIN1/COPS5 3xTg-AD mouse brain, with CD11b as a marker of inflammation (Figure 1A’,D’), GFAP as an astroglial marker (Figure 1B’), and β-amyloid as a marker of extracellular plaque deposits (Figure 1C’). Compared to the wild-type mouse brain (Figure 1A–D), these markers showed strong immunoreactivity in the neocortex (Ctx) and hippocampus (Figure 1A–C). The mean distribution areas of these markers were quantified in each section from both experimental groups with Pixcavator 4.0 software (Figure 1D). Compared to control mice, CD11b immunoreactivity in APP/BIN1/COPS5 3xTg-AD mice increased approximately three-fold in the cortex, thalamus, and hypothalamus (Figure 1A,A’,D,D’), GFAP-immunoreactivity increased two-fold in the hippocampus (Figure 1B,B’), and many β-amyloid plaques were observed only in the APP/BIN1/COPS5 3xTg-AD mouse brain (Figure 1C,C’). Astroglial (GFAP-positive, +) and microglial immune responses (CD11b+) are strong in areas of the brain vulnerable to neurodegeneration; the correlation between these markers indicates the reliability of this model. Taken together, these data revealed that APP/BIN1/COPS5 3xTg-AD mice exhibit an extensive pattern of neuroinflammation that correlated with acute neurodegenerative pathologies, highlighted by increased Aβ deposition, substantial neuroinflammation, and cell death in the neocortex and hippocampus.

### 2.2. AD-Related Gene Expression Is Regulated in APP/BIN1/COPS5 3xTg-AD Mice

*PSEN1* and *PSEN2* are genes that encode the major component of γ-secretase, which is responsible for the sequential proteolytic cleavage of amyloid precursor proteins and the production of β-amyloid peptides. Since alterations in *PSEN2* expression may be a risk factor for AD, we examined *PSEN1* and *PSEN2* expression in hippocampal samples from wild-type and APP/BIN1/COPS5 3xTg-AD mice. PSEN1 mRNA levels were not significantly different between wild-type and APP/BIN1/COPS5 3xTg-AD mice (Figure 2); PSEN2 mRNA levels, however, decreased by approximately 70% in these mice (Figure 2). The ε4 allele of apolipoprotein E (APOE) is the strongest genetic risk factor for late-onset AD [36]. We therefore analyzed APOE mRNA levels in the hippocampus of wild-type and APP/BIN1/COPS5 3xTg-AD mice; APOE expression in APP/BIN1/COPS5 3xTg-AD mice decreased significantly by about 65% (*p* < 0.05) (Figure 2). These findings suggest that APP/BIN1/COPS5 3xTg-AD mice accurately reproduce changes in gene expression associated with AD [37], and are a good preclinical model for studying this neurodegenerative disorder.

As inflammation is a key mechanism in AD, we examined the expression of tumor necrosis factor alpha (TNFα) and interleukin-6 (IL-6) in the hippocampus of wild-type and APP/BIN1/COPS5 3xTg-AD mice. In transgenic mice, TNFα expression increased modestly, but this was not significant (Figure 3A). IL-6 expression, however, was nearly twenty times higher in APP/BIN1/COPS5 3xTg-AD mice than in wild-type animals (*p* < 0.05) (Figure 3A). These data suggest that the APP/BIN1/COPS5 3xTg-AD mouse model replicates the influence of IL-6 on the inflammatory cascade in patients with AD.

Increased nitric oxide synthase 3 (NOS3) expression is associated with cortical neuronal death in AD [38,39]. NOS3 expression slightly increased in APP/BIN1/COPS5 3xTg-AD mice (Figure 3B). Cyclooxygenase-2 (COX-2), an enzyme implicated in inflammatory processes and neuronal activity, is upregulated in the AD brain [40,41]. In the current study, we detected a slight increase in COX-2 expression in the APP/BIN1/COPS5 3xTg-AD mouse hippocampus (Figure 3B), confirming that APP/BIN1/COPS5 3xTg-AD mice are an accurate model for studying AD-like pathology.

### 2.3. Sirtuin Activity and Expression Are Regulated in APP/BIN1/COPS5 3xTg-AD Mice

In recent years, there has been increasing evidence demonstrating the involvement of several epigenetic mechanisms in AD. Sirtuin proteins, together with deacetylase and ADP-ribosyltransferase activity, may provide beneficial effects against age-related disorders [42]. SIRT activity decreased significantly by 15% in the APP/BIN1/COPS5 3xTg-AD mouse hippocampus compared to wild-type controls (*p* < 0.05) (Figure 4A). Moreover, SIRT1 mRNA levels decreased by approximately 50% in AD transgenic mice (Figure 4B); there were no differences in SIRT2 mRNA expression between both experimental groups (data not shown).

### 2.4. HDAC Activity and Expression Are Regulated in APP/BIN1/COPS5 3xTg-AD Mice

HDAC inhibitors have the potential to treat neurological disorders, such as AD [43,44]. We therefore analyzed HDAC activity and HDAC3 mRNA levels in wild-type and APP/BIN1/COPS5 3xTg-AD mice; HDAC activity and HDAC3 mRNA expression increased by approximately 40% in the hippocampus from APP/BIN1/COPS5 3xTg-AD mice (Figure 5A,B). These findings indicate that APP/BIN1/COPS5 3xTg-AD mice are a useful animal model for studying the contribution of epigenetics to AD.

## 3. Discussion

The difficulty of obtaining AD brain tissues presents a major obstacle in AD research. Therefore, mouse models that better mimic human AD-related pathology are critical for the evaluation of therapeutic strategies against this neurodegenerative disorder. In the current study, APP/BIN1/COPS5 3xTg-AD mice showed important brain degenerative hallmarks, such as neuroinflammation, Aβ deposition, and cell death. With respect to the genes that form the basis for the generation of APP/BIN1/COPS5 transgenic mice, mice overexpressing human APP, COPS5, and BIN1 [15] show severe deficits in learning and memory [13,45,46].

Brain-derived neurotrophic factor (BDNF) regulates neuronal development, differentiation, and survival by protecting against tau-related neurodegeneration [47] and is critical to brain health [48,49]. However, deficient BDNF activity underlies neurodegeneration in AD, although exactly how BDNF participates in AD pathology remains unclear. BDNF expression is decreased in hippocampal and cortical neurons from 21-month APP/BIN1/COPS5 3xTg-AD mice [50], and is strongly reduced in buffy coat samples from patients with dementia or PD than in healthy control subjects [51]. In the current study, our findings revealed that the expression levels of two important genes in AD-like disease, *PSEN2* and *APOE*, are altered in the APP/BIN1/COPS5 3xTg-AD mouse brain. Alterations in PSEN2 expression may be a risk factor for AD [52]. Natural compounds that enhance brain apoE levels are potentially therapeutic, given that APOE mRNA levels are regulated by nuclear receptors, retinoid X receptors (RXRs), and hepatic X receptors (LXRs) [53]. Oral administration of the RXR agonist bexarotene, for example, increases apoE levels, reduces Aβ deposition, and improves cognitive functions in APP/PS1 mice [54]. Clinically, four weeks of bexarotene treatment does not reduce brain amyloid burden; the AD patient sample size (*n* = 20, randomized to receive treatment or placebo), however, was small (ClinicalTrials.gov identifier NCT01782742) [55]. RXR heterodimers, including thyroid hormone receptor (TR)/RXR, retinoid acid receptor (RAR)/RXR or peroxisome proliferator-activated receptor γ (PPARγ)/RXR, regulate apoE brain levels, which may serve as targets for the development of anti-AD therapeutics [2,23,56,57]. Our data show that APOE is downregulated in APP/BIN1/COPS5 3xTg-AD mice, suggesting that these mice are suitable for studying the effect of drugs that regulate APOE function.

Neuroinflammation is critical to the progression of AD pathogenesis. IL-6 levels are increased in AD patients [58,59] and are linked to diffuse plaques, representing the early stage of plaque formation [59]. Aβ induces IL-6 expression in microglia and astrocyte cell cultures [59]; IL-6 also induces AD-type tau phosphorylation [60]. TNFα, a critical mediator of the inflammatory response, is secreted by astrocytes and microglia in the brain [59]. TNFα is one of the most well-defined cytokines during AD pathogenesis and is correlated with cognitive decline, neuronal toxicity, and cerebral apoptosis [59]. Increased TNFα expression is found in several regions in the brains of patients with AD [61]. Aβ induces microglial activation and TNFα release, and plays a deleterious role in inducing neuronal death in 3xTg-AD mice [59,62]. In this study, we found a slight increase in TNFα expression and a strong increase in IL-6 levels in the hippocampus of APP/BIN1/COPS5 3xTg-AD mice compared to wild-type controls. As mentioned earlier, NOS3 is related to apoptosis and COX-2 is associated with inflammation [38,39,40,41]. In the present study, transgenic mice showed altered NOS3 and COX-2 expression. However, differences in NOS3, COX-2, and TNFα expression between wild-type and APP/BIN1/COPS5 3xTg-AD mice were not statistically significant. This may be because of the number of mice (*n* = 4 per group) used in the study. Our findings suggest that this model is useful for studying apoptosis and inflammation-related gene expression that influence AD pathology; however, further studies using a higher number of animals are needed. COX-2 inhibition is a proposed strategy for treating patients with AD [63]; non-steroidal anti-inflammatory drugs that selectively inhibit COX-2 reduce the risk of developing AD in a normal-ageing population [64]. Triple-transgenic AD mice may therefore be a valuable preclinical model for testing the effect of drugs that target COX-2 production.

DNA methylation and chromatin remodeling/histone modifications are important epigenetic mechanisms and are promising approaches for the management of AD [65,66]. Global DNA methylation is downregulated in the hippocampus [27] and peripheral blood mononuclear cells in patients with AD [67]. Global DNA methylation is also reduced in the brain, liver, and serum of APP/BIN1/COPS5 3xTg-AD mice; moreover, DNMT3a expression is lower in the APP/BIN1/COPS5 3xTg-AD brain than in wild-type mice [21]. Previously, our group showed that global DNA methylation is decreased in buffy coat samples from patients with AD and other types of dementia, such as vascular or mixed dementia [21,51]. DNMT3a expression was also reduced in buffy coat samples from patients with different neurodegenerative disorders, including AD [51]. These date indicate that DNA methylation is an important biomarker for the diagnosis of neurodegenerative disorders, including AD. There is, furthermore, an age-related increase in DNA hydroxymethylation in the brains of patients with AD [26,56]. In an earlier study, we found that DNA methylation levels improve after treatment with AtreMorine, a bioproduct with potent neuroprotective and dopamine-enhancing capabilities [68]. In APP/BIN1/COPS5 3xTg-AD mice, AtreMorine increases global DNA methylation in brain, liver, and serum samples, with a concomitant increase in DNMT3a expression in the central nervous system [68].

Acetylation is dysregulated in AD and negatively impacts memory and cognition [69,70]. Histone H4 acetylation decreases with age in wild-type mice but increases with age in 3xTg-AD mice [50]. Hippocampal/cortical neurons from young (2–11 month) 3xTg-AD mice respond differently to HDAC inhibition compared to wild-type animals; these transgenic mice are sensitive to the HDAC inhibitor trichostatin A (TSA) early in life (2–11 months) but become insensitive in old age (21 months) [50]. Neurons from non-transgenic mice are TSA-insensitive at young ages but become sensitive at middle-age (8 months) [50]. The contribution of sirtuins in AD has become increasingly clear; SIRT1, for example, regulates APP processing, neuroinflammation, neurodegeneration, and mitochondrial dysfunction in AD [71]. The injection of SIRT1 into the hippocampus of p25 transgenic mice protects against neurodegeneration [72]. Moreover, SIRT1 overexpression reduces tau acetylation and attenuates tau pathology in tau P301S transgenic mice [73]. SIRT1 overexpression, furthermore, reduces AD pathology and decreases the formation of β-amyloid protein and senile plaques in APPSWE/ind (J20) and APPswe/PS1M146L (APP/PS1) mice [71]. Therefore, since SIRT1 overexpression is protective against AD [72], enhancing SIRT1 enzyme activity is an attractive target for investigating novel treatment options in AD [74]. One of the most potent activators of SIRT is resveratrol, a natural chemical found in grapes and red wine that increases SIRT1 activation over ten-fold [74,75]. A variety of synthetic resveratrol derivatives produce lower toxicity and higher potency in the HEK293 cell line and *Saccharomyces cerevisiae* [75]. Several synthetic SIRT activators (e.g., SRT1720, SRT2104, 1,4-dihydropyridine, or UBCS039) activate different SIRTs [57]. SRT1720 and SRT2104 activate SIRT1, 1,4-dihydropyridine activates SIRTs 1–3, and UBCS039 activates SIRT 5 and SIRT6 [57]. In the present study, SIRT activity and SIRT mRNA levels were lower in APP/BIN1/COPS5 3xTg-AD than in wild-type mice; the transgenic AD mouse model used in this study is appropriate for exploring SIRT regulatory mechanisms and SIRT-based therapies for AD. Sirtuin concentrations in saliva can be used for the non-invasive diagnosis of AD in patients of advanced age [76]. SIRT1 concentrations, measured by surface plasmon resonance, western blot, and enzyme-linked immunosorbent assays are substantially lower in AD patients than in non-AD patients; this suggests that SIRT1 is a predictive marker during the early stages of AD [77]. SIRT1, SIRT3, and SIRT6 blood circulatory levels are significantly decreased in AD [78]. Our research group previously showed the importance of SIRT2-related genophenotypes and their implications for AD susceptibility and clinical treatment response [79].

HDAC3 overexpression in the hippocampus of 6-month-old APP/PS1 mice increases Aβ levels, activates microglia, and decreases dendritic spine density [3]. HDAC3 inhibition, however, reverses AD-related pathologies in APP (hAPP695, Swedish mutation), PS1 (PSEN1, M146V), and tau (hTau-4R0N-P301L) mutant mice [31]. HDAC inhibitors, initially used as anti-cancer drugs, may be neuroprotective as they enhance synaptic plasticity and learning and memory in patients with AD [80]. In clinical trials with patients with AD, HDAC inhibitors such as valproate (VPA), suberoylanilide hydroxamic acid (SAHA; Vorinostat), TSA, and sodium butyrate improve memory and reduce cognitive deficits and endogenous Aβ production [44]. The data in the present study support published evidence that HDAC3 inhibition reverses AD-related pathologies in vitro and in vivo, including the 3xTg-AD mice model [31].

There have been promising findings related to blood-based epigenetic biomarkers in AD-based research. However, inconsistencies in research design, technology, the platforms used for biomarker testing, and statistical analysis methodologies have impeded further progress. In fact, relatively few findings have been independently replicated across different research groups [81]. This study, therefore, demonstrates that the APP/BIN1/COPS5 3xTg-AD mouse model is a useful tool for studying novel epigenetic biomarkers for the diagnosis and treatment of AD, with the potential for enhancing translatability.

## 4. Materials and Methods

### 4.1. Animal Model

All experimental procedures were performed in accordance with the European Community Law (86/609/EEC), European Union Directive 2016/63/EU, and the Spanish Royal Decree (R.D. 1201/2005). Study procedures were reviewed and approved by the Ethics Committee of the International Center of Neuroscience and Genomic Medicine.

Wild-type C57BL/6 (*n* = 4) and APP/BIN1/COPS5 3xTg-AD (*n* = 4) male mice were bred from colonies originally donated by Dr. Madepalli Lakshmana (Departments of Immunology and Nano-Medicine, Florida International University, Miami, FL, USA). These mice overexpress the Swedish mutation of human APP, bridging integrator 1 (BIN1), and COP9 constitutive photomorphogenic homolog subunit 5 (COPS5). To distinguish the APP/BIN1/COPS5 model from the well-known 3xTg-AD mouse model (B6;129-Tg(APPSwe,tauP301L)1Lfa-Psen1TM1Mpm/Mmjax), APP/BIN1/COPS5 3xTg-AD refers to the transgenic mice used in the current study. The APP/BIN1/COPS5 3xTg-AD mice were generated at the Sylvester Comprehensive Cancer Center Transgenic Core Facility (University of Miami), using C57BL/6 mice as the background strain. DNA constructs and procedures for the generation of transgenic mice have been previously described [12,45,82]. Nine-month-old mice were housed in an air-conditioned room (22 ± 0.5 °C and 50–60% humidity) under a 12-h light/12-h dark cycle with access to standard laboratory food and water ad libitum. Four animals per experimental group were used.

Mice were genotyped by the polymerase chain reaction (PCR) using genomic DNA from tail samples as templates. Genomic DNA was extracted using a High Pure PCR Template Preparation Kit (Roche, Basel, Switzerland), according to the manufacturer’s specifications. An UltraRun Long Range PCR Kit (Qiagen, Venlo, The Netherlands) and Phire Hot Start II DNA Polymerase (ThermoFisher) were used for *APP* and *BIN*/*COPS5* amplification, respectively. The primers used are shown in Table 1 and PCR conditions in Table 2. IL-2 primers were used as controls for DNA extraction and amplification.

### 4.2. Tissue Collection and Preparation

Wild-type C57BL/6J or APP/BIN1/COPS5 3xTg-AD male mice (*n* = 4 per group; 8–11 weeks-old) were anesthetized with diethyl ether (Panreac, Darmstadt, Germany), and transcardially perfused with phosphate-buffered saline (PBS, pH 7.4). Brains were collected, washed briefly in PBS, and the hemispheres separated. The left hemispheres were fixed in 4% paraformaldehyde in 0.1 M phosphate buffer (pH 7.4) and stored at 4 °C for 48 h. Hippocampi were isolated from the right hemispheres and placed into RNA later solution (Qiagen) for RNA extraction or stored at −80 °C for nuclear extraction.

For immunohistochemistry, the left hemispheres were cryoprotected in 30% sucrose solution in 0.1 M phosphate buffer, and cryosectioned. Parallel series of transverse sections (40–50 µm) were obtained with a cryostat (Starlet 2212, Bright; Luton, UK) and mounted on Superfrost Plus (Menzel-Gläser, Braunschweig, Germany) slides.

### 4.3. Immunofluorescence

Brain sections were rinsed twice for 10 min each in PBS at pH 7.4, and then incubated in nonspecific binding blocking solution (0.1 M PBS containing 0.2% Tween-20 and 15% normal goat serum; Dako; Agilent Technologies, Inc., Santa Clara, CA, USA) for 1 h at room temperature (RT). The sections were then incubated overnight at 4 °C with primary antibodies (see Table 3), diluted in blocking buffer. Negative primary controls included tissues treated with blocking solution but without the addition of primary antibodies. The sections were then washed in PBS and incubated with Alexa-488- or Alexa-594-conjugated secondary antibodies (ThermoFisher Scientific, Waltham, MA, USA), diluted 1:500 in blocking buffer, for 2 h, at RT. The tissues were then washed three times with PBS, counterstained with 4′,6-diamidino-2-phenylindole (DAPI) for 15 min, and coverslipped with Vectashield antifade mounting medium (Vector Labs, Burlingame, CA, USA). Immunostained images were captured with a Leica DM6 B upright microscope (Leica Microsystems, Northbrook, IL, USA) and Leica Application Suite X (LAS X) software version 3.5.5. For each tissue section, area/pixel analysis software (Pixcavator 4) was used to quantify the number of pixels inside the outer boundary of each cell body; this aided quantification of the density of immunofluorescence cell markers relative to background.

### 4.4. Quantitative Real Time RT-PCR (qPCR)

#### 4.4.1. RNA Extraction

Total RNA from hippocampus was extracted with the RNeasy Mini Kit (Qiagen) following the manufacturer’s instructions. The quality and concentration of RNA was measured with microplate spectrophotometer (Epoch, BioTek instruments). Only RNA samples with 260/280 and 260/230 ratios above 1.8 were used in this study.

Purified RNAs (400 ng) were retrotranscribed into DNA according to the specifications of the High Capacity cDNA Reverse Transcription Kit (Thermo Fisher Scientific, Waltham, MA, USA), under the following thermocycling conditions: 25 °C (10 min), 37 °C (120 min), and 85 °C (5 min).

Gene expression was quantified by qPCR using the StepOne Plus Real Time PCR system (Thermo Fisher Scientific, Waltham, MA, USA), following manufacturer’s instructions. Each PCR reaction was performed in duplicate with the TaqMan Gene Expression Master Mix (Thermo Fisher) and specific TaqMan probes (Thermo Fisher) (Table 4). Relative quantification was performed using the comparative CT method [27] with StepOne Plus Real Time PCR software and are expressed as fold induction with respect to healthy samples. Data were normalized to mouse S18 (Table 2) mRNA levels. Data are shown as mean ± S.E.M.

#### 4.4.2. Nuclear Protein Extraction

Nuclear proteins from hippocampal samples were extracted using an EpiQuik Nuclear extraction kit (EpiGentek), according to the manufacturer’s specifications. Protein concentrations were quantified with the Pierce Coomassie (Bradford) Protein Assay Kit (Life Technology, Carlsbad, CA, USA). Absorbances were recorded at 595 nm in an Epoch Microplate Spectrophotometer (BioTek Instruments, Winooski, VT, USA).

#### 4.4.3. Quantification of Sirtuin and HDAC Activity

Sirtuin and HDAC activities were measured by a colorimetric Sirtuin Activity/Inhibition kit (EpiGentek, New York, NY, USA) or HDAC Activity/Inhibition kit (EpiGentek, New York, NY, USA), respectively, following the manufacturer’s instructions. Briefly, nuclear protein extract (50 ng) was transferred to wells that contained an acetylated histone-derived substrate. The plate was then incubated at 37 °C for 90 min, after which the wells were rinsed with 1× wash buffer, and capture and detection antibodies added. The amount of deacetylated product, proportional to SIRT and HDAC enzyme activities, was measured at 450 nm in an Epoch Microplate Reader. Absorbance at 655 nm was used as the reference.

### 4.5. Statistical Analysis

Levene’s test and the Shapiro–Wilk test were used to test for the equality of variances and normality, respectively. Statistical significance was determined with unpaired *t* tests (SPSS software Version 22.0, IBM, Armonk, NY, USA). Data are expressed as mean ± S.E.M, and *p* < 0.05 was considered statistically significant.

## 5. Conclusions

In this study, we evaluated the potential of the APP/BIN1/COPS5 triple-transgenic mouse model for epigenetics-related AD research. It is important to note, however, that our findings are specific to male APP/BIN1/COPS5 3xTg-AD mice. These mice exhibited neuroinflammation and Aβ deposition, which are hallmarks of AD. Furthermore, the expression of three AD-related genes (*PSEN1*, *PSEN2* and *APOE*) was altered. SIRT and HDAC activity and expression were also altered. These data suggest that the APP/BIN1/COPS5 3xTg-AD mouse model is reliable for developing and testing novel epigenetic biomarkers or epidrugs against AD pathology.

## 6. Limitations of the Study

The APP/BIN1/COPS5 mouse model of AD used in this study addresses the early-onset neurodegenerative consequences of Aβ deposition. We did not conduct behavioral tests with this model because the main aim of our study was to investigate a pathogenic role for epigenetics and other biomarkers in the male APP/BIN1/COPS5-3xTg mouse model of AD. Since over 65% of AD cases occur in females, it is important to study AD pathogenesis in female mice. However, female APP/BIN1/COPS5 mice were not used in the current study because we detected a dopaminergic component, exclusively, in the brains of these mice (unpublished data). Our future goal is to investigate the contribution of the dopaminergic system in female APP/BIN1/COPS5 3xTg-AD mice, incorporating molecular and behavioral data; this will have important implications for future drug discovery and clinical trials for AD in terms of targeted sex-specific responses to AD therapeutics. A further limitation of this study is that we did not include a comparison between APP/BIN1/COPS5 mice and the well-known 3xTg-AD and/or 5xFAD transgenic mouse models of AD. However, our future work will address this limitation through behavioral testing, the assessment of cognitive function, and molecular analyses in male and female APP/BIN1/COPS5- and 5xFAD mice.

## Figures and Tables

**Figure 1 ijms-23-02446-f001:**
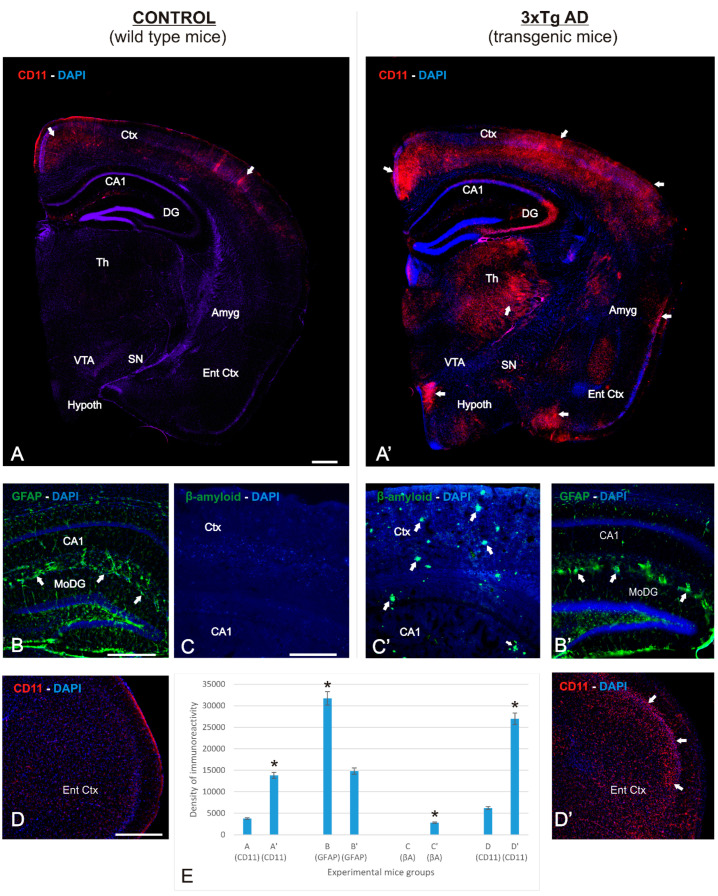
Immunolocalization of various markers in the APP/BIN1/COPS5 3xTg-AD mouse brain. Photomicrographs of different brain regions from wild-type (*n* = 4) and APP/BIN1/COPS5 3xTg-AD (*n* = 4) mice showing histopathological lesions identified via immunohistochemistry. ((**A**–**D**), (**A’**–**D’**)) Transverse sections of right half-brain (**A**,**A’**), dentate gyrus (**B**,**B’**), hippocampus (**C**,**C’**), and entorhinal cortex brain levels of wild-type mice (**A**–**C**) and APP/BIN1/COPS5 3xTg-AD mice (**A’**–**C’**), showing strong differences in expression density. (**A**,**A’**) Hemi-brain sections showing a basal level of the inflammatory marker CD11b in wild-type mice, contrasting with a highly-differentiated agglomeration pattern of immunoreactivity in the APP/BIN1/COPS5 3xTg-AD mouse brain (white arrows). (**B**,**B’**) High magnification of the dentate gyrus showing a marked reduction of GFAP-immunoreactive cells in the APP/BIN1/COPS5 3xTg-AD mouse brain, forming reactive clusters ((**B’**); white arrows) that are absent in wild-type control sections (**B**). (**C**,**C’**) A high density of β-amyloid plaques (arrows in (**C’**)) is found in the APP/BIN1/COPS5 3xTg-AD mouse brain compared to controls (**C**), demonstrating severe neurodegeneration in transgenic mice. (**D**,**D’**) High magnification of the entorhinal cortex showing stronger CD11b immunoreactivity in the APP/BIN1/COPS5 3xTg-AD mouse brain than in controls (**D’**). (**E**) Quantification of data indicating the average percentage of immunoreactive markers in the experimental groups. * *p* < 0.05. Scale bars, 100 μm. Amyg, amygdala; βA, β-amyloid; CA1, hippocampal formation; CD11b, inflammatory marker; Ctx, cortex; DAPI, nuclear marker; DG, dentate gyrus; GFAP, glial fibrillary acidic protein; Th, thalamus; Ent Ctx, entorhinal cortex; Hypoth, hypothalamus; SN, substantia nigra; VTA, ventral tegmental area.

**Figure 2 ijms-23-02446-f002:**
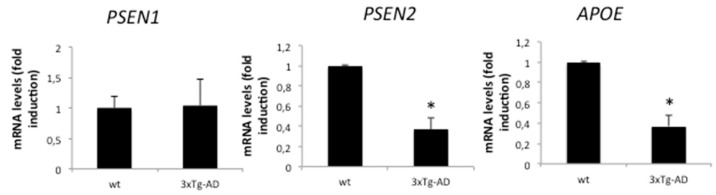
Changes in gene expression in APP/BIN1/COPS5 3xTg-AD mice. PSEN1, PSEN2, and APOE mRNA levels were analyzed in wild-type (*n* = 4) and APP/BIN1/COPS5 3xTg-AD (*n* = 4) mice. Data are expressed as mean ± S.E.M. and as fold-changes compared to mRNA levels in wild-type mice; * *p* < 0.05. APOE, apolipoprotein E, PSEN, presenilin.

**Figure 3 ijms-23-02446-f003:**
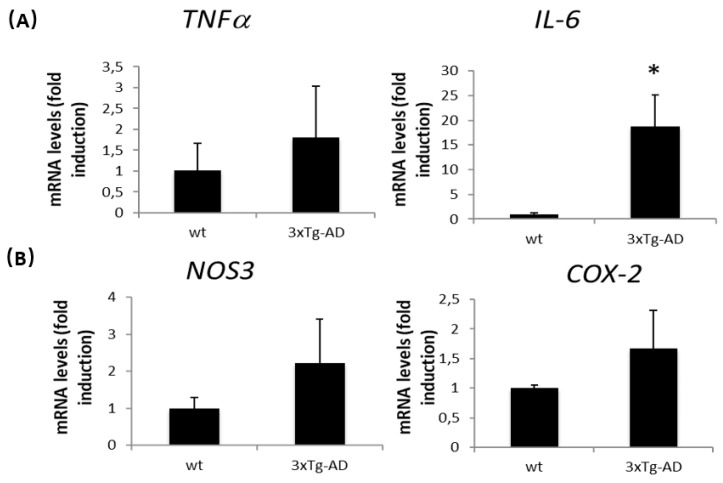
Inflammation and apoptosis-related gene expression in APP/BIN1/COPS5 3xTg-AD mice. TNFα (**A**), IL-6 (**A**), NOS3 (**B**), and COX-2 (**B**) mRNA levels were analyzed in wild-type (*n* = 4) and APP/BIN1/COPS5 3xTg-AD (*n* = 4) mice. Data are expressed as mean ± S.E.M. and as fold-changes compared to mRNA levels in wild-type mice; * *p* < 0.05.

**Figure 4 ijms-23-02446-f004:**
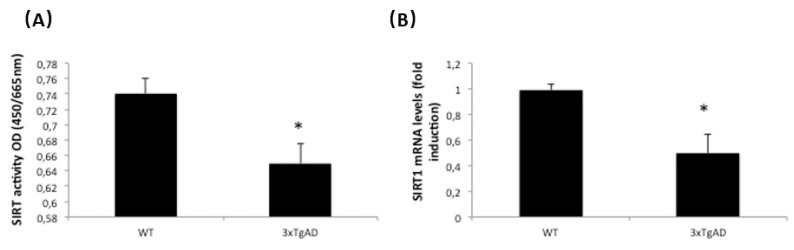
Sirtuins are regulated in APP/BIN1/COPS5 3xTg-AD mice. (**A**) Sirtuin activity was measured as indicated under *Materials and Methods*. (**B**) SIRT1 mRNA levels were measured by qPCR in samples from both wild-type (*n* = 4) and APP/BIN1/COPS5 3xTg-AD (*n* = 4) mice. Data are expressed as mean ± S.E.M; * *p* < 0.05. OD, optical density, SIRT, sirtuin.

**Figure 5 ijms-23-02446-f005:**
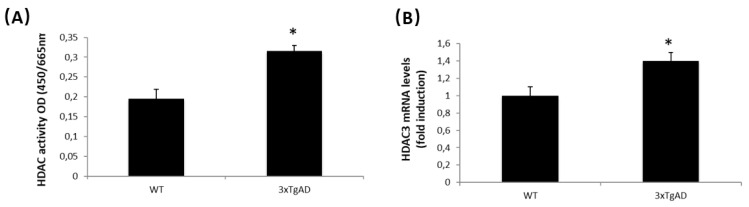
HDACs are regulated in APP/BIN1/COPS5 3xTg-AD mice. (**A**) HDAC activity was measured as indicated under *Materials and Methods*. (**B**) HDAC3 mRNA levels were measured by qPCR in samples from wild-type (*n* = 4) and APP/BIN1/COPS5 3xTg-AD (*n* = 4) mice. Data are expressed as mean ± S.E.M; * *p* < 0.05.

**Table 1 ijms-23-02446-t001:** List of primers used for genotyping.

Primer	Sequence
*APP* forward	AGG ACT GAC CAC TCG ACC AG
*APP* reverse	CGG GGG TCT AGT TCT GCA T
*BIN/COPS5* forward	GAC TAC AAA GAC CAT GAC GGT
*BIN* reverse	CAG GTT AGT TTG AGC TAC GAG
*COPS5* reverse	CCA CCC GAT TGC ATT TTC AAG
*IL-2* forward	CTA GGC CAC AGA ATT GAA AGA TCT
*IL-2* reverse	GTA GGT GGA AAT TCT AGC ATC ATC C

**Table 2 ijms-23-02446-t002:** PCR conditions for mice genotyping.

	Temperature	Time	Cycles
*APP* PCR			
Denaturation	93 °C	3 min	1
Denaturation	93 °C	15 s	40
Annealing	56 °C	30 s	
Extension	68 °C	1 min	1
*BIN/COPS5* PCR			
Denaturation	98 °C	30 s	1
Denaturation	98 °C	5 s	40
Annealing	52 °C	5 s	
Extension	72 °C	15 s	1
Extension	72 °C	1 min	1

**Table 3 ijms-23-02446-t003:** List of antibodies used for immunohistochemistry.

Antibody	Species	Clonality	Supplier	Product Number	Ref.
CD11b	Rat	Monoclonal	ThermoFisher	14-0112-82	[83]
β-amyloid	Rabbit	Monoclonal	ThermoFisher	MA5-35187	[82]
GFAP	Mouse	Monoclonal	ThermoFisher	MA5-12023	[84]

**Table 4 ijms-23-02446-t004:** List of TaqMan probes.

GENE	ID
PSEN1	Mm05001104_m1
PSEN2	Mm00440405_m1
NOS3	Mm0045217_m1
COX-2	Mm0329438_g1
TNFα	Mm0044258_m1
IL-6	Mm00446190_m1
SIRT1	Mm0168521_m1
SIRT2	Mm01492014_m1
HDAC3	Mm0515816_m1
S18	Mm03929990_g1

## Data Availability

Data supporting the reported results are available on request from the corresponding author.

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
