# Peer review of "Epigenetic Studies in the Male APP/BIN1/COPS5 Triple-Transgenic Mouse Model of Alzheimer’s Disease"

_ijms, 2022, doi:10.3390/ijms23052446_

Round 1

Reviewer 1 Report

Martínez-Iglesias et al. investigated the role of epigenetics and other biomarkers in AD using APP/BIN1/COPS5 mice. The study shows altered expression of some AD-related genes, along with altered activity and expression of SIRT and HDAC. The study concludes that the APP/BIN1/COPS5 mouse model is reliable for developing and testing of novel epigenetic biomarkers or epidrugs against neurodegenerative diseases (AD pathology). The study is somewhat interesting; however, it lacks sufficient data and validation to support the conclusion. The study needs to be significantly improved to be acceptable for publication. Specific comments are below.

  • Authors should be aware that "3xTg-AD" mostly refers to the well-known mouse model 3xTg-AD [B6;129-Tg(APPSwe,tauP301L)1Lfa Psen1TM1Mpm/Mmjax] of Alzheimer’s disease (examples; PMID: 30488653, PMID: 12895417), not to the "APP/BIN1/COPS5" model. It may be misleading and authors need to clarify this up front in the manuscript.
  • Introduction, 3rd paragraph, authors cited 79 & 80 which are not actually provided in the list of references. However, APP/BIN1/COPS5 model doesn’t seem to be a well-validated model of AD. Why did authors use this model (APP/BIN1/COPS5) instead of the well-known models for example 3xTg-AD and/or 5xFAD?
  • To claim that this mouse model is a useful tool to study AD, authors should verify and compare these results with other well-studied AD models in this study.
  • How did authors verify that the mouse model developed cognitive impairment and memory deficits pertaining to AD? No behavioral data are provided. Authors should verify this.
  • Why did authors use C57BL/6 mice as controls? What is the actual background mouse strain used for APP/BIN1/COPS5 development? Authors should use the actual background mouse as controls, not just any wild type.
  • Why did authors study only male mice? Authors should be aware that about two-thirds of AD case appears to be in females. It is crucial to study both male and female mice in the study when it comes to studying AD. Authors are encouraged to include female mice in the study and investigate the sex difference.
  • Authors are encouraged to update the Alzheimer’s disease facts and figures in the introduction with the most recent references.

Author Response

The points raised by the reviewer have been addressed as follows (reviewer’ comments are italicized); “Track Changes” has been used to reflect modifications to the revised manuscript.

REVIEWER 1

Comment 1:

Authors should be aware that "3xTg-AD" mostly refers to the well-known mouse model 3xTg-AD [B6;129-Tg(APPSwe,tauP301L)1Lfa Psen1TM1Mpm/Mmjax] of Alzheimer’s disease (examples; PMID: 30488653, PMID: 12895417), not to the "APP/BIN1/COPS5" model. It may be misleading and authors need to clarify this up front in the manuscript.

Response:

Thank you to the reviewer. To distinguish the APP/BIN1/COPS5 mouse model from the well-known 3xTg-AD model [B6;129-Tg(APPSwe,tauP301L)1Lfa Psen1TM1Mpm/Mmjax], we have introduced the term "APP/BIN1/COPS5 3xTg-AD” to specifically refer to the transgenic mice used in our study. This information has been updated, wherever appropriate, in the revised manuscript.

Comment 2:

Introduction, 3rd paragraph, authors cited 79 & 80 which are not actually provided in the list of references. However, APP/BIN1/COPS5 model doesn’t seem to be a well-validated model of AD. Why did authors use this model (APP/BIN1/COPS5) instead of the well-known models for example 3xTg-AD and/or 5xFAD?

Response:

We appreciate this comment by the reviewer. The missing references 80 and 81 have now been added to the revised manuscript. Although the APP/BIN1/COPS5 3xTg-AD mouse model is not as well-known as other transgenic mouse models of AD, we believe that, based on recently published data and concerns related to the translational reliability of animal models of AD, this model may become a well-validated and alternative model of AD. The APP/BIN1/COPS5 3xTg-AD model recapitulates a wide spectrum of neuropathological hallmarks similar to those seen in patients with AD. Presently, no single model develops the full spectrum of AD pathologies, including neuronal loss. A major aim of the present study was to contribute to, and help expand, the number of transgenic mouse models for AD research. In this way, other in vivo models of AD can be used for investigating potential therapeutics against this disease.

Comment 3:

To claim that this mouse model is a useful tool to study AD, authors should verify and compare these results with other well-studied AD models in this study.

Response:

Thank you to the reviewer. Additional information comparing our results to those in other models of AD had been added to the Discussion section in the revised manuscript. In the final paragraph on page 8, and in the second paragraph on page 9, we discuss the roles of SIRT1 and HDAC3 overexpression in other well-studied AD models, respectively. We also point out the neuroprotective effect of HDAC3 inhibition in three different transgenic mouse models of AD.

Comment 4:

How did authors verify that the mouse model developed cognitive impairment and memory deficits pertaining to AD? No behavioral data are provided. Authors should verify this.

Response:

We did not provide behavioral data in the present study because spatial learning and memory skills in APP/BIN1/COPS5 3xTg-AD mice have been verified, as previously described [12,80,81]. APP/BIN1/COPS5 3xTg-AD mice exhibit severe learning and memory deficits, when assessed in the T maze, Barne’s maze, and fear conditioning paradigms of learning and memory.

Comment 5:

Why did authors use C57BL/6 mice as controls? What is the actual background mouse strain used for APP/BIN1/COPS5 development? Authors should use the actual background mouse as controls, not just any wild type.

Response:

Thank you to the reviewer. We have elaborated on the use of C57BL/6 mice as controls in the Materials and Methods section (4.1. Animal model) of the revised manuscript. The background mouse strain is C57BL/6. Linearized APP, BIN1 and COPS5 constructs were first microinjected into the pro-nuclei of fertilized C57BL/6 mouse eggs, and then re-implanted in the oviduct of pseudo-pregnant recipient C57BL/6 mice. To expand the colony, positive founder mice were then backcrossed with native C57BL/6 mice (please see References 80 and 81).

Comment 6:

Why did authors study only male mice? Authors should be aware that about two-thirds of AD case appears to be in females. It is crucial to study both male and female mice in the study when it comes to studying AD. Authors are encouraged to include female mice in the study and investigate the sex difference.

Response:

We appreciate this comment by the reviewer, and agree that it is important to include female mice when studying AD. The major aims in this study were to examine a pathogenic role for epigenetics and other biomarkers in AD. In this pilot experiment, male APP/BIN1/COPS5 3xTg-AD mice were used because we had made an important, unpublished, discovery in the female APP/BIN1/COPS5 mouse brain, that points to dopaminergic dysfunction.

This initial experiment included a) untreated male and female APP/BIN1/COPS5 3xTg-AD mice, and b) male and female APP/BIN1/COPS5 mice treated with a bioactive product (AtreMorine) similar to L-DOPA that increases brain dopamine levels.

The brains of these mice were then assessed via Western blotting for tyrosine hydroxylase (TH) protein expression, normalized to the housekeeping protein GAPDH.

We found:

1) In untreated APP/BIN1/COPS5 3xTg-AD mice, baseline TH protein levels were substantially lower in female than in male mice.

2) Enhancing dopamine synthesis and release via AtreMorine supplementation restored TH levels in female APP/BIN1/COPS5 3xTg-AD mice.

3) TH levels were not affected in AtreMorine-supplemented non-transgenic (WT) male mice.

These data suggest that alterations in dopamine signaling may contribute to the development of AD pathology in female APP/BIN1/COPS5 3xTg-AD mice. In the present study, we therefore chose to use only male mice to avoid the experimental confound of a deregulated dopaminergic system.

Our future study objectives, however, are to examine epigenetic mechanisms and the expression of other biomarkers in female APP/BIN1/COPS5 3xTg-AD mice. This would address the course of AD pathology, by examining features such as locomotion, goal-directed behavior, anxiety, and cognitive abilities, and would represent a therapeutic target for female patients with AD.

In the Conclusions section of the revised manuscript, we have clarified that the data in the present study are specific to male APP/BIN1/COPS5 3xTg-AD mice.

Comment 7:

Authors are encouraged to update the Alzheimer’s disease facts and figures in the introduction with the most recent references.

Response:

In the revised manuscript, we have updated the Alzheimer’s disease facts and figures with the most current information available (please see paragraph 1 of the Introduction section)

Olaia, please check that references stated here, match the numbering in the References section of the manuscript.

 [79] W. J., J. A., L. M., B. A., L. Y., B. J., P. S., K. E., M. I., K. D., Pivotal role of the RanBP9-cofilin pathway in Aβ-induced apoptosis and neurodegeneration. Cell Death Differ. 19 (2012) 1413-23.

[80] W. R., W. H., C. I., X. S., L. M., COPS5 protein overexpression increases amyloid plaque burden, decreases spinophilin-immunoreactive puncta, and exacerbates learning and memory deficits in the mouse brain. J Biol Chem. 290 (2015) 9299-309.

Reviewer 2 Report

The lack of efficacy of existing therapies for Alzheimer´s disease is attributed to diagnosis at late stages of the disease and limited knowledge of biomarkers and molecular mechanisms of the pathology.  Thus, the aim of this study was to investigate a pathogenic role for epigenetics and biomarkers in a triple-transgenic mouse model.  The authors demonstrated that the transgenic mice exhibited neuroinflammation and amyloid-β deposition, the expression of genes altered, and activity of sirtuin and HDAC was altered.  The authors conclude that the triple-transgenic mouse model is reliable for developing and testing of novel epigenetic biomarkers or epidrugs against Alzheimer´s disease.  There are some major concerns that must be addressed.

Comments:

In Abstract, Page 1 line 7, “In the 3xTg mouse hippocampus, sirtuin and HDAC3 expression and activity decreased”  However, HDAC3 mRNA expression increased by approximately 40% in the hippocampus in section 2.4 (Page 6).

In Abstract, Page 1 line 9, “and levels of pro-inflammatory (COX-2, TNFα, and IL-6) and apoptotic (NOS3) markers increased.”  This sentence is misleading, because the expression of COX-2, TNFα, and NOS3 was not significant.

In Conclusions, Page 12 line 1, “The expression of numerous AD-related genes was, furthermore, altered.”  The results showed that three genes were significantly different between wild-type and the triple-transgenic mice.

Page 6, 4th line from the bottom, “HDAC activity and HDAC3 mRNA expression increased by approximately 40% in the hippocampus from both experimental groups (Fig. 5A, 5B).” Please revise this sentence.

Author Response

The points raised by the reviewer have been addressed as follows (reviewer’ comments are italicized); “Track Changes” has been used to reflect modifications to the revised manuscript.

REVIEWER 2

Comment 1:

In Abstract, Page 1 line 7, “In the 3xTg mouse hippocampus, sirtuin and HDAC3 expression and activity decreased”  However, HDAC3 mRNA expression increased by approximately 40% in the hippocampus in section 2.4 (Page 6)

Response:

Thank you to the reviewer. In the Abstract, this sentence has been revised as “In the APP/BIN1/COPS5 3xTg-AD mouse hippocampus, sirtuin expression and activity decreased, HDAC3 expression and activity increased.”

Comment 2:

In Abstract, Page 1 line 9, “and levels of pro-inflammatory (COX-2, TNFα, and IL-6) and apoptotic (NOS3) markers increased.”  This sentence is misleading, because the expression of COX-2, TNFα, and NOS3 was not significant.

Response:

In the Abstract, we have corrected this sentence as follows: “levels of pro-inflammatory COX-2 and TNFα and apoptotic (NOS3) markers increased slightly, but these were non-significant.”

Comment 3:

In Conclusions, Page 12 line 1, “The expression of numerous AD-related genes was, furthermore, altered.”  The results showed that three genes were significantly different between wild-type and the triple-transgenic mice.

Response:

Thank you to the reviewer. The Conclusions section of the revised manuscript now includes information that states that the expression of three specific genes increased. This sentence now reads “The expression of three AD-related genes (PSEN1, PSEN2 and APOE) was, furthermore, altered.”

Comment 4:

Page 6, 4th line from the bottom, “HDAC activity and HDAC3 mRNA expression increased by approximately 40% in the hippocampus from both experimental groups (Fig. 5A, 5B).” Please revise this sentence.

Response:

This sentence has now been revised as “HDAC activity and HDAC3 mRNA expression increased by approximately 40% in the hippocampus from APP/BIN1/COPS5 3xTg-AD mice (Fig. 5A, 5B).”

We sincerely hope that we have addressed the comments to the satisfaction of the reviewer.

Sincerely,

Dr Olaia Martínez-Iglesias

Department of Medical Epigenetics,

EuroEspes Biomedical Research Center

Round 2

Reviewer 1 Report

Authors responded to the comments, without adding any further experiments/data. This reviewer does not find the clarifications made by the authors sufficient for this study to be acceptable for publication in its current form.

-Authors added the references 80 and 81, but those studies do not seem to have used the same mouse model. Searching on Pubmed with the key word “APP/BIN1/COPS5” does not show any papers published with this model either, which further reinforces that it may not be a well-validated model for studying AD.

-Authors responded that …..”Learning and memory skills in APP/BIN1/COPS5 3xTg-AD mice have been verified, as previously described [12, 80, 81].” However, previous study (Ref 12) does not seem to have done any behavioral studies. And other studies (Ref 80 and 81) seem to have used different mouse models. Authors are encouraged to do the behavioral studies.

-Regarding background strain, only the study (ref 81) seems to have mentioned “The linearized thy-1.2-FLAG-COPS5 construct was microinjected into the pro-nuclei of fertilized C57BL/6 mouse eggs…” to generate FLAG-COPS5 mice, but not the APP/BIN1/COPS5. It needs to be clear and justified, and the control experiments should be done using the actual background strain.

-However, Authors are encouraged to expand their study by including appropriate well-known models (e.g., 3xTg-AD, 5xFAD) to validate their findings. This will significantly improve the current study.

-When the major aims are to examine the pathogenesis of AD, using both sex of animals is crucial to make a conclusion. Thus, authors are encouraged to perform/include all experiments/data for both male and female to make a more meaningful study.

Author Response

Thank you for your email. We appreciate the criticism given by the reviewer and hope that our revised manuscript “Epigenetic studies in the male APP/BIN1/COPS5 triple-transgenic mouse model of Alzheimer’s disease” will be acceptable for publication as a Research Article in International Journal of Molecular Sciences (Special Issue: Physiological or Pathological Molecular Alterations in Brain Aging).

The points raised by the reviewer have been addressed as follows (reviewers’ comments are italicized); “Track Changes” has been used to reflect modifications to the revised manuscript.

REVIEWER 1

Comment 1:

Authors added the references 80 and 81, but those studies do not seem to have used the same mouse model. Searching on Pubmed with the key word “APP/BIN1/COPS5” does not show any papers published with this model either, which further reinforces that it may not be a well-validated model for studying AD.

Response:

Thank you to the reviewer. We acknowledge that a validated model is important for studying AD. In our study, we have referenced the association between the APP/BIN1/COPS5 3xTg mouse model and AD in three other publications (Carrera et al. 2017; Martínez-Iglesias et al. 2020a; Martínez-Iglesias et al. 2020b).

            Our published paper (Carrera et al. 2017) compared the development of extracellular Aβ deposits between APP/PS1 and APP/BIN1/COPS5 transgenic mice. Male and female APP/BIN1/COPS5 mice developed extracellular Aβ deposits earlier than APP/PS1 animals. The severity of Aβ pathology was also more severe and earlier in onset in APP/BIN1/COPS5 than in APP/PS1 mice. These data showed that the APP/BIN1/COPS5 mouse model is useful for studying AD pathology. Since trigenic mouse models of AD are more pathogenic than bigenic mice models of AD, our objective was to analyze the epigenetic impact of AD in a trigenic (APP/BIN1/COPS5) mouse model of AD.

            In the next paper (Martínez-Iglesias et al. 2020), we used APP/BIN1/COPS5 mice to analyze

the effects of a bioactive product (AtreMorine) that increases brain dopamine levels. Global DNA methylation (5-methylcytosine, 5mC) and DNA methyltransferase 3a (DNMT3a) levels were significantly lower in the brain from 6-9-week-old APP/BIN1/COPS5 mice than controls.

In a complementary manuscript (Martínez-Iglesias et al. 2020b), we confirmed that global DNA methylation is reduced in the brain, liver and serum from APP/BIN1/COPS5 3xTg-AD mice; moreover, DNMT3a expression was lower in the APP/BIN1/COPS5 3xTg-AD brain than in wild-type mice.

            In the present study, our primary aim was to expand our investigation into the epigenetic effects in the brain in APP/BIN1/COPS5 male mice with AD, but to also supplement the initial histopathological data in Carrera et al. (2017). We examined SIRT1 and HDAC3 expression and activity; PSEN1, PSEN2, APOE, COX-2, IL-6, TNFα and NOS3 expression; and CD11b and β-amyloid immunoreactivity.

References:

Carrera I.; Novoa L.; Teijido O.; Sampedro C.; Seoane S.; Lakshmana M. K. & Cacabelos R. (2017). Comparative characterization of transgenic mouse models of Alzheimer´s disease. J Genom Med Pharmacogenom. 2, 331-337.

Martínez-Iglesias O., Naidoo V., Carril J C., Carrera I., Corzo L, Rodriguez S, et al. (2020a). AtreMorine Treatment Regulates DNA Methylation in Neurodegenerative Disorders: Epigenetic and Pharmacogenetic Studies. Current Pharmacogenomics and Personalized Medicine. 17, 159-171.

Martinez-Iglesias O.; Carrera I.; Carril J.C.; Fernández-Novoa L.; Cacabelos N.; Cacabelos R. DNA methylation in Neurodegenerative and Cerebrovascular Disorders. (2020b). Int J Mol Sci. 21(6), 2220.

Comment 2:

Authors responded that ”Learning and memory skills in APP/BIN1/COPS5 3xTg-AD mice have been verified, as previously described [12, 80, 81].” However, previous study (Ref 12) does not seem to have done any behavioral studies. And other studies (Ref 80 and 81) seem to have used different mouse models. Authors are encouraged to do the behavioral studies.

Response:

We apologize to the reviewer for being unclear and wish to clarify our previous statement that “learning and memory skills in APP/BIN1/COPS5 3xTg-AD mice have been verified, as previously described”.

Our intention was to state that we had added References 80 and 81 (References 45 and 83, respectively, in the revised version) to the text to support behavioral studies relative to the genes incorporated into APP/BIN1/COPS5 3xTg-AD mice. We have now corrected and incorporated this information into the first paragraph of the Discussion section in the revised manuscript.

We did not perform behavioral studies because our goal in this manuscript was not to study behavior, but to evaluate epigenetic changes in relation to pathological hallmarks in a male AD mouse model. Figure 1 was included in the manuscript to demonstrate the extent of AD-related brain degeneration.

Comment 3:

Regarding background strain, only the study (ref 81) seems to have mentioned “The linearized thy-1.2-FLAG-COPS5 construct was microinjected into the pro-nuclei of fertilized C57BL/6 mouse eggs…” to generate FLAG-COPS5 mice, but not the APP/BIN1/COPS5. It needs to be clear and justified, and the control experiments should be done using the actual background strain.

Response:

Previously cited references 80 and 81 (References 45 and 83, respectively, in the revised version) were added to the manuscript to additionally support the evidence of the background strain used (C57BL/6) in the generation of this experimental mice model, as detailed in the published article (Ref. 12). This strain was also used for cross-breeding as established by the transgenic guidelines in animal experimental manipulation, also referred to in the published article (Ref. 12).

Comment 4:

-However, Authors are encouraged to expand their study by including appropriate well-known models (e.g., 3xTg-AD, 5xFAD) to validate their findings. This will significantly improve the current study.

Response:

It is not possible for us to include a comparison of the 5xFAD or a different 3xTg-AD model in the present manuscript, because we do not have these animals and will need to procure them. We appreciate this comment, however, and agree that these are important experiments to perform in the future.

Comment 5:

When the major aims are to examine the pathogenesis of AD, using both sex of animals is crucial to make a conclusion. Thus, authors are encouraged to perform/include all experiments/data for both male and female to make a more meaningful study.

Response:

Thank you to the reviewer. The aim of this study was to investigate a pathogenic role for epigenetics and other biomarkers in the male APP/BIN1/COPS5-3xTg mouse model of AD (Lines 6-7, Abstract) – we have now included “male” in the corrected title of the manuscript. We did not use female APP/BIN1/COPS5 mice in the current study because: 1) we detected a dopaminergic component, exclusively, in the brain of female APP/BIN1/COPS5 mice; this information has been added to the Materials and Methods section (4.1 Animal model) in the revised manuscript. 2) Females show hormonal diversity and exhibit differences in circadian rhythms and behavior compared to males. We, therefore, opted to use only males in this investigation to eliminate these confounding variables.

            Although over 65% of AD cases appear to be in females, it is also important to study AD pathogenesis in males. A recent study (Abd-Elrahman et al. 2020), using transgenic AD mice, showed that AD therapy targeting the neuronal glutamate receptor mGluR5 was effective in reversing the disease in male mice but not in female mice. This has important implications for future drug discovery and clinical trials for AD in terms of targeted sex-specific responses to AD therapeutics.

Nonetheless, our future aim is to study the contribution of the dopaminergic system in female APP/BIN1/COPS5 3xTg-AD mice, with molecular and behavioral data.

Our unpublished data show the following:

1) In APP/BIN1/COPS5 3xTg mice, baseline levels of tyrosine hydroxylase (TH) are lower in untreated female (lanes 4 and 9) than untreated male mice (lanes 2 and 7).

2) AtreMorine restores TH levels in female APP/BIN1/COPS5 mice (lanes 5 and 10).

3) AtreMorine downregulates TH levels in male APP/BIN1/COPS5 mice (lanes 3 and 8). Our additional (unpublished) Western blot data show that administration of AtreMorine to wild-type (WT) male mice had no effect on TH levels (data not shown).

Fig. 1.1                                                           Fig. 1.2

Fig. 1.1 Western blot data showing TH levels in the midbrain of male and female APP/BIN1/COPS5-3xTg in the presence or absence of AtreMorine (n=8). Fig. 1.2 Immunofluorescence images showing the reduced density of TH-immunoreactive neurons (arrows) in the substantia nigra in female (B) versus male (A) APP/BIN1/COPS5-3xTg mice. AT, AtreMorine; GAPDH; glyceraldehyde-3-phosphate dehydrogenase; MW, molecular weight marker; TH, tyrosine hydroxylase; UT, untreated. Scale bar, 100 µm.

Reference:

Abd-Elrahman KS., Albaker A., de Souza JM., Ribeiro FM., Schlossmacher MG., Tiberi M,. et al. (2020). Aβ oligomers induce pathophysiological mGluR5 signaling in Alzheimer's disease model mice in a sex-selective manner. Sci Signal. 13(662):eabd2494.

We hope that we have addressed the comments to the satisfaction of the reviewer.

Sincerely,

Dr Olaia Martínez-Iglesias

Department of Medical Epigenetics

EuroEspes Biomedical Research Center

Bergondo, 15165

Corunna

Spain

E-mail: epigenetica@euroespes.com

Reviewer 2 Report

This manuscript is improved.

Author Response

Thank you for considering that manuscript has been improved. We appreciate the review given by the reviewer and hope that our revised manuscript “Epigenetic studies in the male APP/BIN1/COPS5 triple-transgenic mouse model of Alzheimer’s disease” will be acceptable for publication as a Research Article in International Journal of Molecular Sciences (Special Issue: Physiological or Pathological Molecular Alterations in Brain Aging).

Sincerely,

Dr Olaia Martínez-Iglesias

Department of Medical Epigenetics

EuroEspes Biomedical Research Center

Bergondo, 15165

Corunna

Spain

Round 3

Reviewer 1 Report

-It is good that the authors clarified in the discussion that they did not perform behavioral studies. Similarly, it would be good to write other limitations as well. Thus, authors are encouraged to write a separate limitation section in the manuscript making all the limitations clear; for examples, limitations of this mouse model, lack of behavioral studies (as they mentioned), lack of verification of these results with other established models, lack of data from female mice, including other limitations authors may find and focusing on the importance of future studies.

-Authors may want to check if they have mentioned the age of the mice in the method section.

-Authors are encouraged to remove “We did not use female APP/BIN1/COPS5 mice in the current study because of a strong dopaminergic component that we detected, exclusively, in the brain of female APP/BIN1/COPS5 mice (unpublished data).” from the Methods section and include in the Limitation section.

-Authors are encouraged to make sure n value (number of animals and/or experiments) is mentioned in all figure legends.

Author Response

Thank you for your email. We appreciate the criticism given by the reviewer and hope that our revised manuscript “Epigenetic studies in the male APP/BIN1/COPS5 triple-transgenic mouse model of Alzheimer’s disease” will be acceptable for publication as a Research Article in International Journal of Molecular Sciences (Special Issue: Physiological or Pathological Molecular Alterations in Brain Aging).

The points raised by the reviewer have been addressed as follows (reviewers’ comments are italicized); “Track Changes” has been used to reflect modifications to the revised manuscript.

Comment 1:

It is good that the authors clarified in the discussion that they did not perform behavioral studies. Similarly, it would be good to write other limitations as well. Thus, authors are encouraged to write a separate limitation section in the manuscript making all the limitations clear; for examples, limitations of this mouse model, lack of behavioral studies (as they mentioned), lack of verification of these results with other established models, lack of data from female mice, including other limitations authors may find and focusing on the importance of future studies.

Response:

Thank you to the reviewer. We have now included a new section “6. Limitations of the study” on page 12 of the revised manuscript. All of the points raised by the reviewer have been addressed in that section of the paper.

Comment 2:

Authors may want to check if they have mentioned the age of the mice in the method section..

Response:

We apologize to the reviewer for this oversight. We have added the age of the mice to the Materials and Methods section (4.1. Animal model) of the revised manuscript.

Comment 3:

Authors are encouraged to remove “We did not use female APP/BIN1/COPS5 mice in the current study because of a strong dopaminergic component that we detected, exclusively, in the brain of female APP/BIN1/COPS5 mice (unpublished data).” from the Methods section and include in the Limitation section.

Response:

Thank you to the reviewer. As suggested, we have removed this sentence from the Methods section, and added it to section “6. Limitations of the study”.

Comment 4:

Authors are encouraged to make sure n value (number of animals and/or experiments) is mentioned in all figure legends.

Response:

Thank you to the reviewer. The number of animals used (n values) has been added to all figure legends.

We hope that we have addressed the comments to the satisfaction of the reviewer.

Sincerely,

Dr Olaia Martínez-Iglesias

Department of Medical Epigenetics

EuroEspes Biomedical Research Center

Bergondo, 15165

Corunna

Spain

E-mail: epigenetica@euroespes.com